# Effect of Heterogeneous Microstructure on Refining Austenite Grain Size in Low Alloy Heavy-Gage Plate

**Shengfu Yuan [1], Zhenjia Xie [2] , Jingliang Wang [2], Longhao Zhu [3], Ling Yan [3], Chengjia Shang [2,3,*] and R. D. K. Misra [4]**

[1]  School of Materials Science and Engineering, University of Science and Technology Beijing, Beijing 100083, China; yuansf_77@163.com
[2]  Colarborative Innovation Center of Steel Technology, University of Science and Technology Beijing, Beijing 100083, China; zjxie@ustb.edu.cn (Z.X.); jlwang@ustb.edu.cn (J.W.)
[3]  State Key Laboratory of Steel for Marina Equipment and Application, Anshan Steel, Anshan 114021, China; zlhdut@163.com (L.Z.); yanling_1101@126.com (L.Y.)
[4]  Laboratory for Excellence in Advanced Steel Research, Department of Metallurgical and Materials Engineering, University of Texas at El Paso, El Paso, TX 79912, USA; dmisra2@utep.edu
*  Correspondence: cjshang@ustb.edu.cn; Fax: +8610-6233-2428

**Abstract:** The present work introduces the role of heterogeneous microstructure in enhancing the nucleation density of reversed austenite. It was found that the novel pre-annealing produced a heterogeneous microstructure consisting of alloying elements-enriched martensite and alloying-depleted intercritical ferrite. The shape of the martensite at the prior austenite grain boundary was equiaxed and acicular at inter-laths. The equiaxed reversed austenite had a K-S orientation with adjacent prior austenite grain, and effectively refined the prior austenite grain that it grew into. The alloying elements-enriched martensite provided additional nucleation sites to form equiaxed reversed austenite at both prior austenite grain boundaries and intragranular inter-lath boundaries during re-austenitization. It was revealed that prior austenite grain size was refined to ~12 μm by pre-annealing and quenching, while it was ~30 μm by conventional quenching. This is a practical way of refining transformation products by refining prior austenite grain size to improve the strength, ductility and low temperature toughness of heavy-gage plate steel.

**Keywords:** alloying element enrichment; heterogeneous microstructure; nucleation site; grain refinement

## 1. Introduction

Low carbon low alloy steels have been widely used in fields of marine engineering, engineering machines, and other structural applications due to their high strength, excellent toughness, good weldability and low cost [1–3]. With the structural components becoming large and the need for cost effectiveness, there is a significant demand for heavy gauge plate steel. However, for low carbon low alloy heavy-gage plate steel manufactured by thermomechanical controlling process (TMCP), austenite grains in the center of the plate gradually become larger due to insufficient reduction in thickness during controlled rolling. This leads to the deterioration of mechanical properties, especially low-temperature toughness [4–6]. Therefore, a heat treatment process should be applied to optimize the mechanical properties of heavy plate.

Austenite grain refinement is well recognized as being of great importance in enhancing mechanical properties of high strength low alloy steels [7–9]. Several efforts have been made towards the refining of austenite grain size. Grange [10] reported that the thermal cycling process between martensite and austenite is helpful in refining austenite grain size by reversed transformation. The effect of the

two-step quenching (TSQ) and one-step quenching (OSQ) heat treatment process on austenite grain size was studied [11], and the results showed that the prior austenite grain size of TSQ heat-treated specimen was 50% finer than OSQ heat treatment. Further studies [12–14] found two morphologies of reversed austenite from martensite or bainite: equiaxed austenite and acicular austenite during thermal cycling process and multi-step heat treatment process. In addition, the mechanism of prior austenite grain refinement was attributed to the enhancement of nucleation density of equiaxed austenite by thermally activated nucleation or static recrystallization [15].

There is a vast literature [13,14,16–21] reporting that reversed austenite nucleates at the prior austenite grain boundary, intersection of multiple-grains, packet boundary, block boundary and cementite. Miyamoto [22] pointed out that austenite preferentially nucleates at the high angle boundary of pearlite-ferrite. Studies [23,24] on the crystallographic characteristics determined that acicular austenite had Kurdjumov–Sachs (K-S) orientation relationship $((111)_{fcc}//(011)_{bcc}, [-101]_{fcc}//[-1-11]_{bcc})$ with the matrix, while equiaxed austenite nucleated at prior austenite grain boundaries grew into the adjacent prior austenite grain at high angle misorientation [23]. This also suggested that the equiaxed austenite was beneficial in refining austenite grain size. In addition, some literatures [25–28] published that cementite was the core of austenite nucleation. However, in this study, the content of carbon is 0.08 wt. %, which was too low to grow to the core of austenite nucleation. In contrast, the cementite was easily decomposed. In this study, a novel pre-annealing process prior to quenching was introduced to increase nucleation density of equiaxed reversed austenite and refine austenite grain size in the core of heavy plate. Microstructure evolution during heat treatment was studied to elucidate the mechanism of austenite grain refinement by pre-annealing and quenching.

## 2. Experimental Material and Procedure

The chemical composition of experimental steel was C 0.08, Si 0.23, Mn 1.21, P 0.011, S 0.002, Ni 1.1, (Cu + Cr + Mo) < 2.0, Nb 0.022, B 0.0012 and Ti 0.014 in weight percent (wt. %). The $A_{C1}$ and $A_{C3}$ temperatures were measured by dilatometry to be 715 and 863 °C, respectively. Three specimens (denoted as 1#, 2#, 3#) with dimensions of 10 mm (rolling direction) × 3 mm (transverse direction) × 10 mm (thickness) were cut from the center of the experimental plate steel which was hot rolled to 100 mm by TMCP, because of inferior mechanical properties of center plate. Heat treatments were carried out as shown in Figure 1. Specimen 1# was reheated to 900 °C for 30 min for complete re-austenitization, and then quenched to room temperature (Figure 1a). To isolate the effect of intercritical annealing, specimen 2# was isothermally held for 30 min at 740 °C followed by water quenching to room temperature (Figure 1b), to distinguish the microstructure after intercritical annealing. In the case of specimen 3#, two-step heat treatment process: intercritical annealing at 740 °C for 30 min followed by quenching and reheating to 900 °C for 30 min followed by quenching, was applied, as shown in Figure 1b.

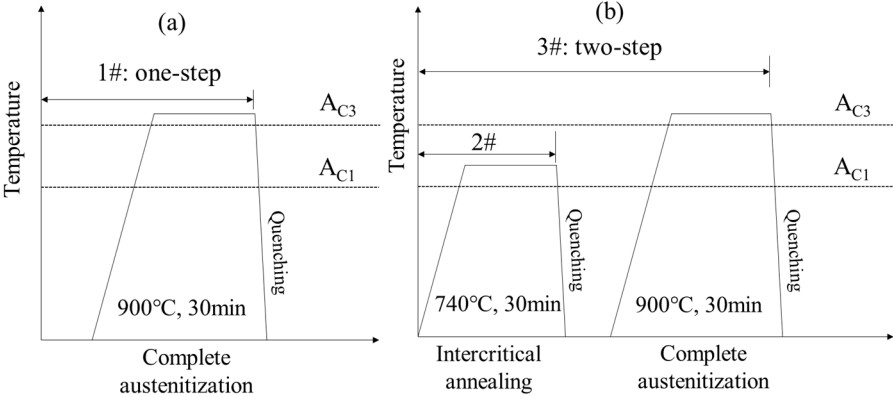

**Figure 1.** Heat treatment scheme of experimental specimens: (**a**) conventional quenching process; and (**b**) intercritical annealing, and intercritical annealing + conventional quenching processes.

The microstructure was characterized by optical microscope (OM) (OLYMPUS, Beijing, China), field emission scanning electron microscope (FEG-SEM; ULTRA 55) and electron backscatter diffraction (EBSD, Oxford Instruments, England, UK). For OM, the specimens were metallographically polished and etched by Picric acid (China electron microscope scientific instrument, China) consisting of 10 mL detergent (Diao Pai made in Nice Group), 2 mL CCl4, 0.2 g NaCl, 4.5 g picric acid and 60 mL deionized water in 50–55 °C water bath in order to observe the equiaxed austenite grain size. The specimens were etched by 4% nital for SEM observation. The local chemical composition in specimen 2# was examined by energy dispersive x-ray spectrometer (EDS) facility available with transmission electron microscope. The austenite transformation vs temperature during reheating was simulated by dilatometer at a heating rate of 0.1 °C/s for the one-step and two-step processes. The specimens for EBSD analysis were ground and electrolytically polished in an electrolyte of 10% perchloric acid, 5% glycerol and 85% ethanol at 15 V for 30 s. The EBSD measurements were performed at a voltage of 20 kv with a step size of 0.1 μm.

## 3. Results and Discussion

### 3.1. Refining Prior Austenite Grain

Figure 2 shows the effect of two-step heat treatment on the refinement of austenite grain size in the core of heavy-gage plate processed by TMCP. The austenite grain size of hot rolled specimen was ~105 μm in the core, as presented in Figure 2a, because the cooling rate was lower in the core of the hot rolled plate; the microstructure was composed of bainite with large martensite and austenite constituents (M/A). After conventional one-step quenching of hot rolled plate (specimen 1#), the prior austenite grain was inhomogeneous, the large austenite grain ($\gamma_b$: ~78 μm) was surrounded by small austenite grain ($\gamma_a$: ~29 μm), as shown in Figure 2b. For the specimen 2# (intercritical annealed at 740 °C for 30 min), as shown in Figure 2c, where it can be seen that austenite grain size was as large as that of the hot rolled sample (~106 μm), but there was fresh martensite formed at the prior austenite grain boundary (G-M) and the inter-lath (L-M). The microstructure of specimen 2# was composed of G-M, L-M and intercritical ferrite (I-F); as indicated in Figure 2c, G-M and L-M was transformed from equiaxed reversed austenite (ERA) and distributed at the prior austenite grain boundary, and acicular reversed austenite (ARA) was present at bainite-lath. However, after the two-step heat treatment (specimen 3#), fine and uniform austenite grains were obtained, as shown in Figure 2d. The average grain size was ~12 μm, which was twice as fine as specimen 1#. It can be seen that the novel pre-annealing heat treatment process enhanced the nucleation of reversed austenite and refined the coarse prior austenite grains.

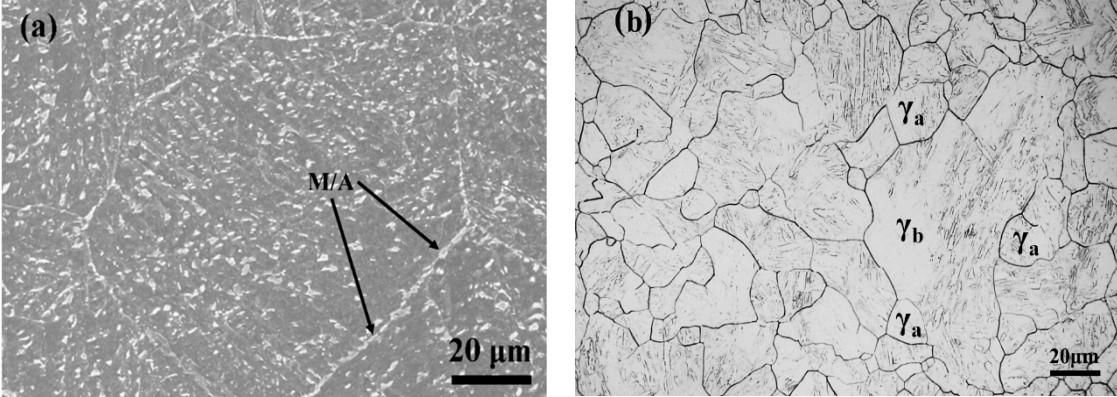

**Figure 2.** *Cont.*

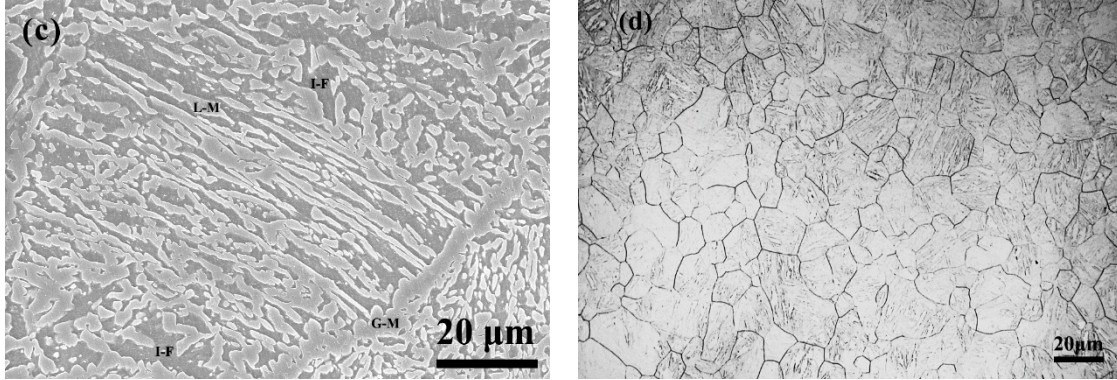

**Figure 2.** SEM images of hot rolled plate steel (**a**) and specimen 2# (**c**) by nital etching; optical microscope images of specimen 1# (**b**) and specimen 3# (**d**) by picric etching; M/A: martensite and austenite constituent; I-F: intercritical ferrite; L-M: martensite distributed at laths; G-M: martensite distributed at prior austenite grain boundary; $\gamma_a$ and $\gamma_b$: austenite grain.

### 3.2. Crystallographic Characteristics of Pre-Annealed Microstructure

The pre-annealed specimen (2#) was investigated by EBSD in order to analyze the crystallographic orientation of fresh martensite at prior austenite grain boundary, as presented in Figure 3. A typical fresh martensite 'b' between two prior austenite grains γ-A and γ–B was selected, as shown in Figure 3a. From the pole figures, Figure 3b–d, it was found that the position of "one" particular Bain group of selected fresh martensite in the (100) pole figure was identical to "one" Bain group of prior austenite grain γ–B, which indicated that the selected fresh martensite held the K-S orientation relationship with the prior austenite grain γ-B, but no orientation relationship with the prior austenite grain γ-A. In addition, from the pole figures, Figure 3b,c, it was found that the fresh martensite between inter-laths had the same Bain groups with intercritical annealed ferrite, which suggested that the fresh martensite kept the K-S orientation relationship with the prior austenite grain. The fresh martensite between the inter-laths was formed by acicular reversed austenite when cooling to room temperature. This illustrated that the acicular reversed austenite held the K-S orientation relationship with the prior austenite grain. Sadovskii [29] showed that it is possible to reconstruct the prior austenite by growth and impingement of acicular austenite during austenitization.

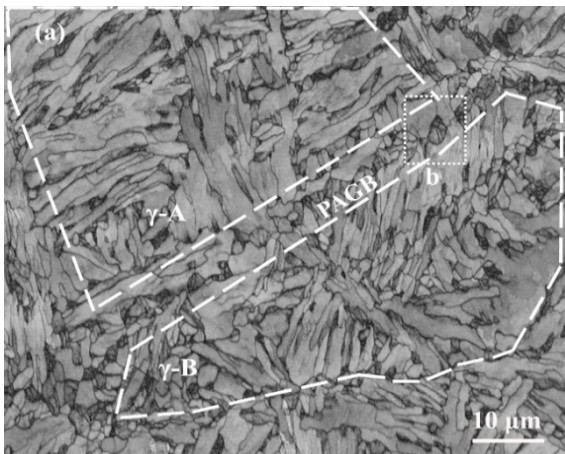

**Figure 3.** *Cont.*

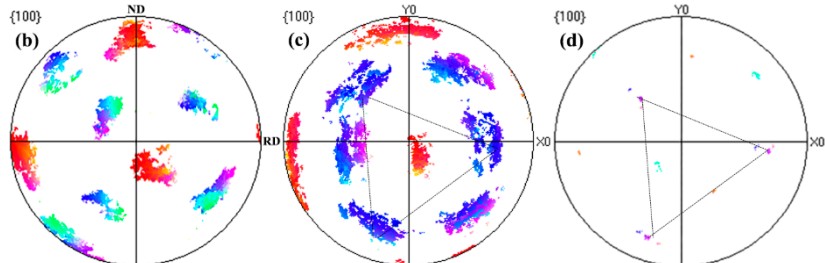

**Figure 3.** (**a**) Band contrast images of 2# specimen; two pole figures (**b**,**c**) corresponding to the prior austenite grain γ-A and γ-B in image (**a**); pole figures (**d**) corresponding to the selected sections of 'b' in image (**a**).

### 3.3. Enrichment of Alloying Elements

The distribution of alloying elements in specimen 2# was examined, as presented in Figure 4. From intercritical ferrite to martensite to intercritical ferrite on the other side, line scanning of alloying elements (Figure 4a) showed that the contents of alloying elements, Mn, Ni, Cu and Cr increased in martensite and decreased in intercritical ferrite (I-F), as shown in Figure 4b. Five-line scanning results are shown in Table 1. It can be seen that the concentration of alloying elements Mn, Ni, Cu and Cr was significantly higher in contrast to their nominal compositions, which formed the heterogeneous microstructure of alloying element enriched martensite and depleted intercritical ferrite. In combination with the image of Figure 2d, the heterogeneous microstructure is related to fine and uniform austenite grains obtained by the two-step heat treatment.

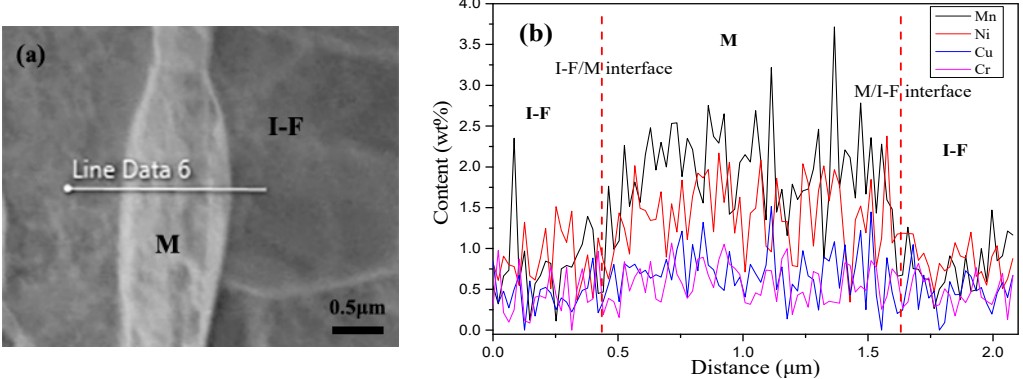

**Figure 4.** Examined schematic image of fresh martensite after intercritical annealing (**a**); examined results of alloying elements distribution by FEG-SEM (**b**).

**Table 1.** Concentration of alloying elements in M for specimen 3#.

| Position | Mn | Ni | Cu | Cr |
|----------|------|------|------|------|
| 1# | 1.89 | 1.59 | 0.68 | 0.51 |
| 2# | 2.08 | 1.67 | 0.85 | 0.68 |
| 3# | 1.81 | 1.56 | 0.72 | 0.56 |
| 4# | 1.95 | 1.57 | 0.79 | 0.59 |
| 5# | 2.09 | 1.74 | 0.82 | 0.65 |
| Mean | 1.96 | 1.63 | 0.77 | 0.60 |

### 3.4. Role of Heterogeneous Microstructure

The role of the heterogeneous microstructure in phase transformation of austenite was studied by thermal simulation (dilatometer) experiments, as shown in Figure 5. The start temperature and finish temperature of austenite transformation were 718 °C and 877 °C, respectively, for the

one-step heat treatment specimen, and for the two-step heat treatment specimen, they were 695 °C and 881 °C, respectively, as presented in Figure 5a,c. We can see that from the plot of the rate of austenite transformation and heating temperature, there was only a peak transformation rate for the one-step heat treatment specimen (Figure 5b), and there were two peak rates for the two-step heat treatment specimen (Figure 5d). This indicated that the alloying elements-enriched fresh martensite had lower transformation temperature and alloying elements-depleted intercritical ferrite had higher transformation temperature.

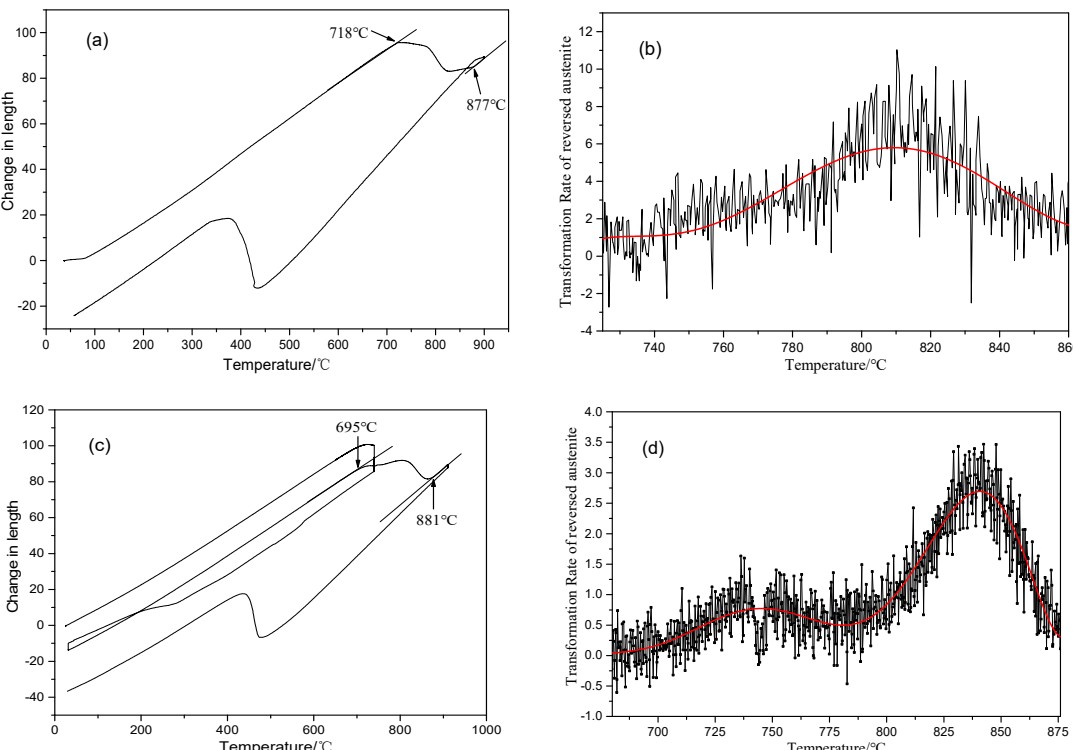

**Figure 5.** Gleeble simulated curves of one-step (**a**) and two-step (**c**) heat treatment; the austenite transformation rate curves of one-step (**b**) and two-step (**d**) heat treatment.

The austenite nucleation during reheating for intercritically annealed specimen can be discussed from the viewpoint of thermodynamics. As is well known, the thermal activated nucleation needs to overcome the energy barrier before the formation of embryo. According to the nucleation theory [30], under ideal state, the relationship of the nucleation energy barrier, interface energy and intrinsic energy difference of fcc phase and bcc phase can be expressed by:

$$\triangle G_0 = 16\pi\gamma^3/(3^*(\triangle g)^2) \tag{1}$$

where $\Delta G_0$ is nucleation energy barrier, $\gamma$ is interface energy, $\Delta g$ is intrinsic energy difference of fcc phase and bcc phase when bcc phase transformed into fcc phase. The nucleation energy barrier is decreased when the interface energy is low (in our case, the interface energy was low because of alloying elements enriched at the interface) and the intrinsic energy difference of fcc phase and bcc phase increased. Thus, the necessary driving energy for nucleation was small, which enhanced the nucleation rate of reversed austenite via thermally activated nucleation.

$$\Delta g = g_{fcc} - g_{bcc} \tag{2}$$

where $g_{fcc}$ or $g_{bcc}$ represent intrinsic energy of $\alpha$ phase or $\gamma$ phase, respectively, at a temperature and at a given chemical composition.

The composition of G-M/L-M and I-F after intercritical annealing at 740 °C was calculated by Thermo-calc 3.0 based on TCFE 7.0 according to the nominal composition (NC). The alloying element contents in G-M/L-M and I-F were 0.15C-0.21Si-2.03Mn-1.87Ni-0.52Cr-0.36Mo-0.80Cu (wt. %) and 0.05C-0.24Si-0.79Mn-0.74Ni-0.43Cr-0.48Mo-0.35Cu (wt. %), respectively. Figure 6 shows the intrinsic energy difference of fcc phase and bcc phase for the nominal composition, martensite (G-M/L-M) and I-F, respectively, during reheating. It can be seen that the intrinsic energy difference of G-M/L-M was higher than NC when the temperature was below 763 °C, and the intrinsic energy difference of I-F was greater than NC as the temperature was greater than 815 °C. Thus, the heterogeneous microstructure can reduce the energy barrier and driving energy for nucleation, and enhance the nucleation of equiaxed reversed austenite.

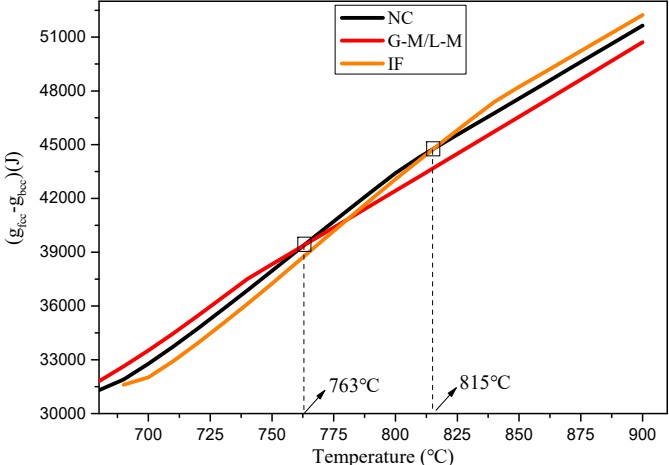

**Figure 6.** The intrinsic energy difference of fcc phase and bcc phase when bcc phase transformed into fcc phase for the compositions of NC, G-M/L-M and I-F calculated by thermo-calc 3.0 based on TCFE 7.

### 3.5. Nucleation Rate of Reversed Austenite

The effect of heterogeneous microstructure on reaustenitization during reheating was studied via step by step experiments. Experiments were designed every 20 °C in the temperature range of 720–860 °C for one-step and two-step heat treatment processes, and the results are shown in Figure 7. Two optical microscope images show the distribution of equiaxed reversed austenite grain (Figure 7a,b) for the one-step and two-step heat treatment processes. It can be seen that the equiaxed reversed austenite grain was large and mainly distributed at prior austenite grain boundary for one-step heat treatment specimen (Figure 7a); however, the fine equiaxed reversed austenite grain obtained by two-step heat treatment was present at the prior austenite grain boundary and intragranular (Figure 7b). The results in Figure 7b showed the heterogeneous microstructure can enhance the nucleation of equiaxed reversed austenite. The number of equiaxed reversed austenite grains for different intercritical annealing temperature were counted, as shown in Figure 7c (note: the statistical grain size was for an area greater than 2 square micro-meters; for the two-step experimental specimens, the grain below 820 °C was too small to count). The austenite grain number for the two-step specimen was higher than for the one-step specimen for every intercritical annealing temperature, which indicated that the nucleation density of equiaxed grain for two-step specimen was higher in contrast to that of the one-step specimen.

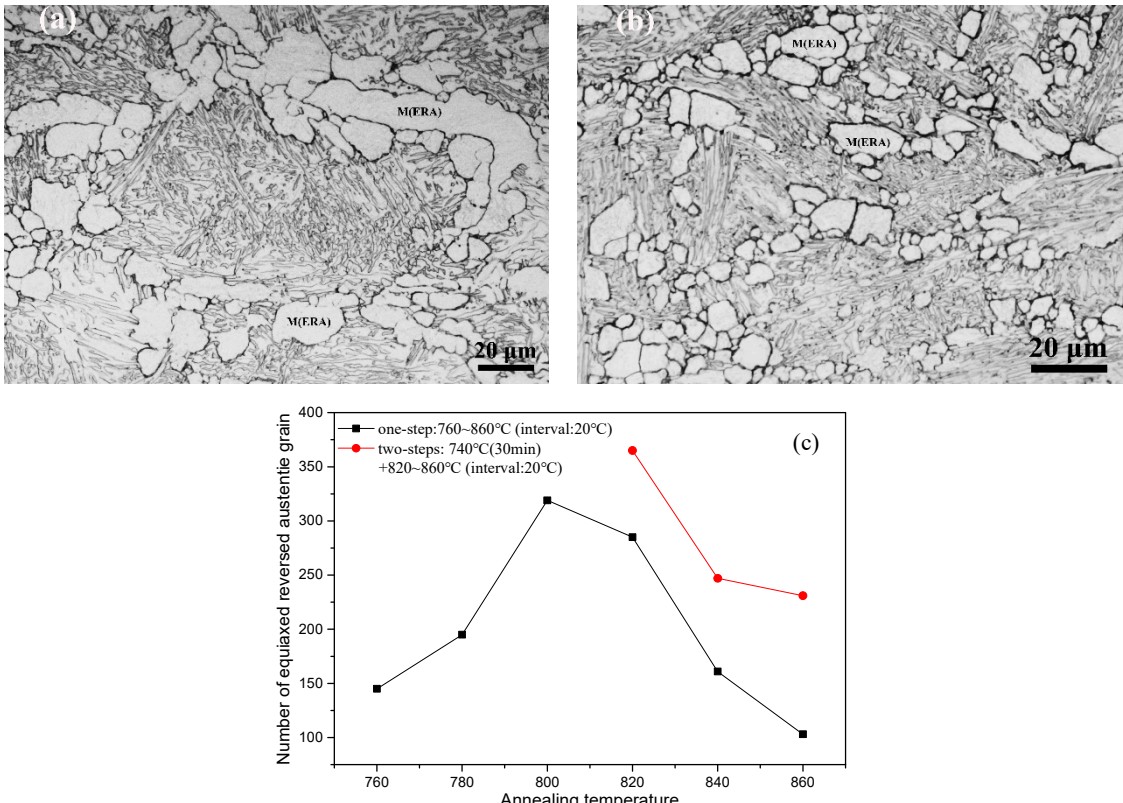

**Figure 7.** The optical microscope images of one-step (**a**) and two-step (**b**) at 820 °C holding 1 min; (**c**) the number of statistically equiaxed reversed austenite grain (within the size of $165 \times 114$ square micro meter).

In a summary, a schematic image of austenitized behavior during one-step and two-step austenitization is proposed, as shown in Figure 8. For the one-step heat treatment specimen, the equiaxed reversed austenite (ERA) grains only occurred at the prior austenite grain boundary and grew fast during austenitization (Figure 8b) to form the austenite grain $\gamma_a$ (Figure 8c), the acicular reversed austenite (ARA) grew and impinged to reconstitute the prior austenite $\gamma_b$ (Figure 8c), as presented in Figure 2b. For the two-step specimen, the heterogeneous microstructure of alloying elements enriched fresh martensite (including G-M and L-M) and depleted intercritical ferrite was obtained after intercritical annealing (Figure 8d), which enhanced the equiaxed reversed austenite grain nucleated at the intragranular and prior austenite grain boundary during reaustenitization (Figure 8e). Finally, fine and uniform austenite grain size was obtained after complete austenitization, (Figures 8f and 2d).

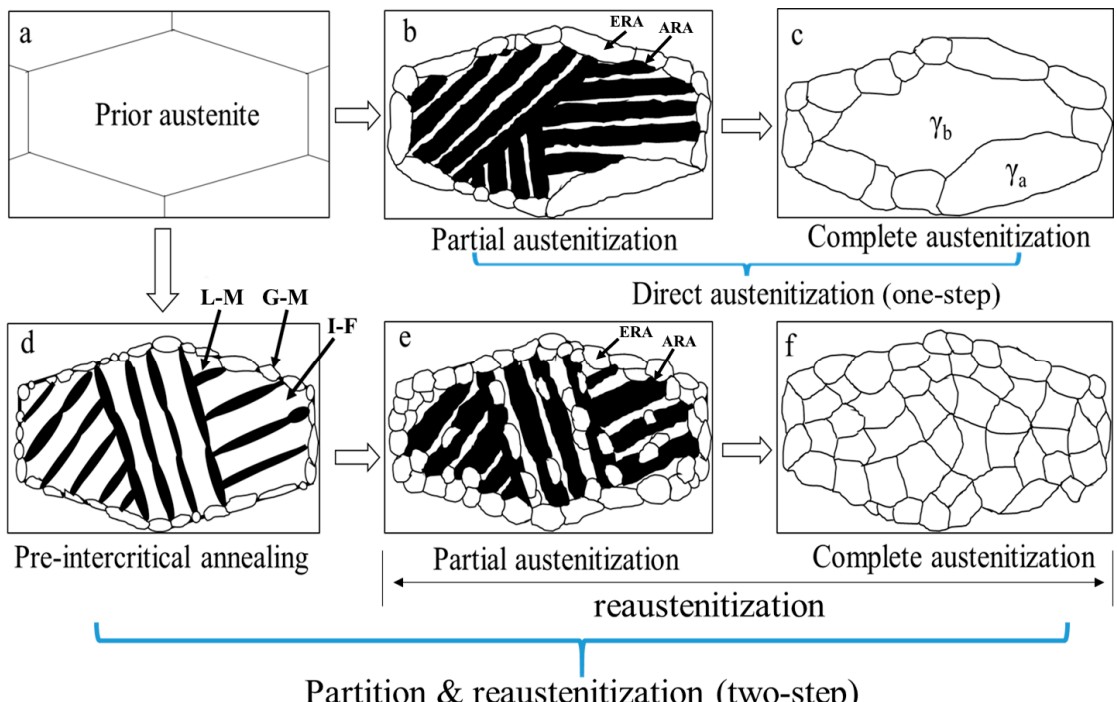

**Figure 8.** Schematic of direct austenitization (**a**–**c**) or partition and reaustenitization (**a**,**d**–**f**) for hot rolled experimental steel. ERA: equiaxed reversed austenite; ARA: acicular reversed austenite.

## 4. Conclusions

In this study, coarse prior austenite grains in the core of heavy plate processed by TMCP were studied by two-step heat treatment, and the conclusions are concluded as follows:

- The heterogeneous microstructures of alloying elements-enriched fresh martensite and -depleted intercritical ferrite were obtained after intercritical annealing. The fresh martensite was distributed at prior austenite grain boundary (G-M) and inter-lath (L-M). G-M was transformed by equiaxed reversed austenite and the L-M was formed by acicular reversed austenite, when the reversed austenite obtained during intercritical annealing was quenched to room temperature.
- The heterogeneous microstructure increased the intrinsic energy difference of fcc phase and bcc phase below 763 °C or above 815 °C, and the interface energy decreased because of alloying element enrichment at the interface, which reduces the nucleation energy barrier according to the nucleation theory under ideal state. Therefore, the nucleation driving energy of equiaxed reversed austenite is less. The heterogeneous microstructure can enhance the equiaxed reversed austenite nucleation at intragranular and prior austenite grain boundary during reheating, which effectively refined the coarse prior austenite grains in the core of hot rolled heavy plate processed by TMCP.
- The prior austenite grains in the core of heavy plate processed by TMCP was very large (~105 µm). The austenite grains were inhomogeneous when the hot rolled specimen was reheated by one-step heat treatment process. However, for the two-step heat treatment process, fine and uniform austenite grain size (~12 µm) was obtained, which was two times finer compared to the one-step heat treatment. An effective way of improving strength, ductility and low temperature toughness in alloy steel is to refine the prior austenite grain size. This study provides a possible way of effectively refining prior austenite grain size.

**Author Contributions:** S.Y.: conceptualization, data curation, formal analysis, investigation, writing—original draft; Z.X.: review & editing; J.W.: formal analysis, review & editing; L.Z.: data curation; L.Y.: dcuration; C.S.: conceptualization, funding acquisition, supervision, review & editing; R.D.K.M.: review & editing. All authors have read and agree to the published version of the manuscript.

**Funding:** This research received no external funding.

**Conflicts of Interest:** The authors declare no conflict of interest.

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
