# Peer review of "Effect of Heterogeneous Microstructure on Refining Austenite Grain Size in Low Alloy Heavy-Gage Plate"

_metals, doi:10.3390/met10010132_

Round 1
Reviewer 1 Report
The authors studied the microstructure of a low-carbon steel subjected to two different heat treatments, one called OSQ (one-step quenching) consisting in an austenitization (AC3 + 50°C) of the sample followed by water quenching and the other called TSQ (two-step quenching), composed of an annealing (AC1 + 50°C) and an austenitization.
The different microstructures observed are perfectly well explained in the figure 8.
The initial microstructure is bainitic i.e composed of bainite with large martensite and austenite constituent, after the OSQ, the authors observe a microstructure composed of large austenitic grains decorated by small austenitic grains.
After annealing, the microstructure is martensitic in the prior austenite grain boundary decorated with a necklace of small austenitic grains. The austenitization treatment makes it possible to obtain an equiaxial austenitic grain structure.
Despite a very interesting metallurgical study of the effect of heat treatment on the microstructure of a low carbon steel, it is a very simple study (which does not exceed the “bachelor level”) with a small scientific interest. The authors have tried to add substance to their article and showing some EBSD results with Kurdjumov-Sachs orientation relationship (KS-OR), an EDX profile on a martensitic lath and some thermocalc’s calculation of driving energy, but all this does not present much interest if one presents them thus. It lacks a systematic characterization of the mechanical properties (Re, Rm, A%) because the interest of an equiaxial structure is not exploited. For all these reasons, I reject this publication.

Author Response
Dear reviewer 1,
Thank you very much for your examination, I have corrected the errors and responded to your questions, please receice and check it.
Best wishes!
Dr. Yuan

Reviewer 2 Report
The manuscript could be accepted for publication after minor revision (corrections to minor methodological errors and text editing).

Author Response
Dear reviewer,
Thank you very much for your careful modification for my paper. Most of them were the errors of words or sentences, I have corrected them according to your requirements in text (in red). And some of your confusing questions were responded in text, because of small questions. Therefore, I did not listed one by one, and please receive and check it. Keep in touch if you have any questions, thanks again!
Best wishes to you!
Dr. Yuan

Reviewer 3 Report
The manuscript describes two step annealing and quenching treatment to refine austenite grain. The microstructure was successfully refined and the mechanism is discussed from the view point of thermodynamics. The two step annealing and quenching process would be of great interest to the readers. The experimental procedure and results are clearly presented. However, the introduction is not sufficient to understand the novelty of the study and I have several concerns with the discussion about nucleation. Overall, I would recommend major revision and re-reviewing before ready for publication. Specific comments are listed below:
In Introduction, the authors say "a novel pre-annealing process prior to quenching was introduced" but what is the novelty of the process is not explained. In addition, I found a number of papers about two step annealing and intercritical annealing by few minutes search by Google Scholar. It may be appropriate to include references regarding them and compare the work with the authors' work to show the novelty of the study. Line 156 to 172, the authors discuss the mechanism of refinement based on the nucleation theory but only the activation energies for nucleation are evaluated. The ratio of nucleation rate can be calculated from the activation energies by using nucleation theory if the other parameters are assumed to be constant. Can the difference in activation energy for nucleation explain the difference of the number of nucleation of austenite? Line 156 to 207, judging from Fig. 7(b), the new austenite grain is nucleated at the grain boundaries and the interfaces, in other words, the nucleation is not homogenous but heterogeneous. I think the authors should discuss the difference of other parameters in nucleation rate including the density of nucleation sites and whether any of them affect the nucleation rate in the heat treatment process.
Here are minor comments:
Line 94-95, "equiaxed reversed (ERA) austenite" should be "equiaxed reversed austenite (ERA)". Figure 4(b), can the enrichment be explained by local equilibrium of ferrite and austenite calculated by TCFE7? Figure 4(b), interface position should be denoted for readers' understanding.
Author Response
Dear reviewer,
Thank you very much for your examination of my work. I have corrected the errors and answered your questions, if you have any questions,please contact me!
Best wishes!
Dr. Yuan

Reviewer 4 Report
Metals-659025: Effect of heterogeneous microstructure on refining austenite grain size in low alloy heavy plate steel
The topic is appropriate and interesting for Metals. In my opinion, the paper is well organized, and the references are adequate. Anyway, some recommendations should be considered:
* The number of references should be higher. Furthermore, it would be advisable to include some papers from the journals of MDPI editorial (Metals, Applied Sciences, Materials, etc.) related to the topic of the manuscript.
* Abstract is not attractive for the reader. Authors should emphasize the interest of this research work in the abstract (e.g. regarding the contribution of the paper, the introduction section is clearer than the abstract).
* Lines 24-32. Several references could be added in these sentences.
* In addition, Conclusion section should include a sentence emphasizing the practical contribution of this research work (not only the scientific contribution).
Author Response
Dear editor
I have modified the abstract and conclusions, and some literatures have been introduced to explain the text. The changed text was in red , please receive and check it, thank you!

Round 2
Reviewer 4 Report
Metals-659025: Effect of heterogeneous microstructure on refining austenite grain size in low alloy heavy plate steel
The paper was improved but some recommendations should still be considered:
* To establish a better fit for the journal Metals, it would be advisable to include some references from such a journal.
* It would be advisable to include the practical usefulness of this paper in both the Abstract and the Introduction sections, as the last sentence of Conclusion section (emphasizing the practical contribution of this research work).
Author Response
Dear refree
I have modified the paper according to your suggestions. Please check again.
